# Separation of Rare Earth Elements in Multistage Extraction Columns in Chromatography Mode: Experimental Study and Mathematical Simulation

**Artak E. Kostanyan** , **Vera V. Belova \***, **Yulia V. Tsareva and Maria M. Petyaeva**

Kurnakov Institute of General and Inorganic Chemistry, Russian Academy of Sciences, 31 Leninskii pr., 119991 Moscow, Russia; kost@igic.ras.ru (A.E.K.); tsareva.juliana@ya.ru (Y.V.T.); maria.mikhailovna1994@yandex.ru (M.M.P.)

\* Correspondence: belova@igic.ras.ru

**Abstract:** The application of liquid–liquid chromatography principles to solvent extraction processes in hydrometallurgy can greatly simplify rare earth metal separation technologies by separating multi-component mixtures in one technological operation. In this study, the chromatographic separation of rare earth elements (REEs) in multistage extraction columns was experimentally studied under conditions of impulse sample injection—single and multiple loading of large volumes of metal salt solution into the installation. The results obtained showed the feasibility of operating sieve plate extraction columns in the liquid–liquid chromatography mode. A closed-loop recycling technology is proposed for the separation of rare earth elements in multistage extraction columns operating in the liquid–liquid chromatography mode. For further development and industrial implementation of this technology, experimental studies should be conducted on intensified multistage extraction columns, such as sectioned columns with agitators and vibrating plate columns. Computer simulation of the chromatographic separation of rare earth elements by closed-loop recycling liquid–liquid chromatography was carried out.

**Keywords:** separation of rare earth elements; liquid–liquid chromatography; solvent extraction; multistage extraction columns; closed-loop recycling chromatography



## 1. Introduction

Solvent extraction methods are widely used in hydrometallurgy to separate metals with similar properties, in particular for the separation and purification of rare earth elements [1–11]. Rare earth elements are widely applied in different fields: electronics, medicine, permanent magnet manufacturing, and catalysis. The separation and purification of individual rare earth elements is challenging due to their remarkably similar chemical properties. At present, in addition to extraction methods, various rare earth separation and recovery methods have been developed, which include chemical precipitation, ion exchange, crystallization, adsorption, chromatography, membrane separation, etc. However, solvent extraction processes are most commonly used to separate rare earth elements in industry. Currently, 2-ethylhexyl phosphinic acid mono-2-ethylhexyl ester (P507) is most often used for the REE separation. For this extraction system, the separation factors of individual REEs have been established, and operational parameters have been developed for the equipment and chemical reagents used [5]. Compared with other extractants, systems with P507 exhibit relatively high separation factors between REEs and good physical characteristics. However, this process has several disadvantages. The separation factors for some elemental pairs (for example, Nd/Pr, Gd/Eu, Lu/Yb) are very small, requiring many equilibrium extraction stages for achieving high purities in the separated products. In addition, the stripping acidity required for the heavier elements is quite high. Cyanex

272 (bis-2,4,4-trimethylpentylphosphinic acid) has for a long time been considered as an alternative extractant to P507 [6]. This phosphinic acid has a lower acid dissociation constant (Ka) than P507, i.e., it is a weaker extractant, of which the stripping processes proceed at a lower aqueous acidity. However, the use of Cyanex 272 is limited due to the poor physical characteristics of this extractant. Subsequently, it was found that in systems involving mixtures of P507 and Cyanex 272, stripping processes proceeded more easily than with P507 alone [9,10]. An improvement in separation factors was noted for some pairs of heavy REEs. Later, a new extractant was released under the name Cyanex 572 (Cytec Industries), specially designed for the separation of rare earth elements, which is a mixture of phosphinic and phosphonic acids (P507 and Cyanex 272) [11]. This extractant provides efficient extraction of heavy rare earth elements simultaneously, with the stripping of these metals using mineral acid solutions with a lower concentration than in the systems with P507. Commercially available extractants such as Cyanex 572 are immediately available for industrial use and have shown significant potential for optimizing and improving the separation and concentration of individual rare earth elements [11].

The application of chromatography principles to solvent extraction processes can greatly improve industrial separation technologies. In solvent extraction and in liquid–liquid chromatography, referred to as "counter-current chromatography" (CCC) [12–16] and "centrifugal partition chromatography" (CPC) [17–19], the separation of substances is based on their different solubility in two liquid phases. Countercurrent chromatography differs from traditional liquid chromatography in that the so-called stationary phase in a CCC device is not fixed on a solid carrier but is kept mobile in the device due to centrifugal forces. The undoubted advantage of CCC separation methods over the methods of solvent extraction is the possibility of separating multicomponent mixtures in one technological operation (within a single process stage), which greatly simplifies technological schemes. In addition, since the organic phase containing the extractant is not removed from the chromatographic device, the consumption of chemicals is significantly reduced, which allows the use of more selective, but more expensive, chemicals.

Currently, CCC separations are mainly used for analytical and preparative purposes for the separation and purification of herbal pharmaceutical products [12–19]. CCC devices in the form of a tube wound in one or several layers around the drum of a planetary centrifuge (hydrodynamic devices), or in the form of a cascade of chambers on the surface of a cylinder, or on a pack of discs mounted on the shaft of a conventional centrifuge (hydrostatic devices), are difficult to scale up. In addition to the complexity of the equipment, the existing CCC devices are characterized by low productivity. Thus, the disadvantage of CCC processes is the complexity of instrumentation and the associated low productivity, which prevents their use in large-capacity industrial production, for example, in hydrometallurgy. The productivity of extraction apparatuses used in industry (extraction columns and cascades of series-connected mixer–settlers) exceeds the productivity of chromatographic apparatuses by several orders of magnitude. Therefore, for creating high-performance devices that can be used in industry for the isolation of chemical compounds from solutions, their separation and purification by methods of liquid–liquid chromatography is an urgent task. To overcome the above disadvantages of CCC apparatuses, high-performance industrial-scale CCC separations can be developed using conventional liquid extraction equipment [20–26]. In [21], the chromatographic behavior of europium, gadolinium, dysprosium, samarium, terbium, and yttrium in the cascade consisting of 70 centrifugal mixer–settler extractors operating in the isocratic elution mode was experimentally studied. A mixture of 30 vol.% Cyanex 572 + 10 vol.% tributylphosphate in a hydrocarbon diluent was used as the stationary phase, and aqueous nitric acid was used as the mobile phase. It was experimentally shown that modified centrifugal mixer–settlers (the stationary organic phase recirculates between the mixing and separating chambers) can operate in chromatography mode and be used for large-scale separation of rare earth elements [21].

Multicomponent two-phase solvent systems used in liquid–liquid chromatography can be an alternative to conventional extraction systems [27–29]. At present, several multicomponent two-phase solvent systems are applied in countercurrent chromatography, for example, ARIZONA solvent systems (ethyl acetate–water–heptane–methanol) [30], HEMWat systems (hexane–ethyl acetate–methanol–water) [31], and others. Multicomponent solvent systems differ from extraction systems in the possibility of varying the number of solvent components and their ratio, which allows achieving appropriate distribution coefficients of substances. However, countercurrent chromatography is mainly used to separate natural products in physical extraction. The addition of extractants to multicomponent two-phase solvent systems makes it possible to create a new class of extraction systems for separating metals [28,29]. Previously, we studied the distribution of REEs in ternary solvent systems containing di(2-ethylhexyl)phosphoric acid [32,33], binary extractants of various compositions and their mixtures [28,34,35], and Cyanex 572 [29,36]. It was established that the hexane–isopropanol–water (2:1:1.5) ternary system with Cyanex 572 is promising for the separation of metals of heavy-light and medium groups. The extraction of REEs with Cyanex 572 proceeds by the cation-exchange mechanism at a relatively low acidity of the aqueous phase, and the stripping of metals can be performed using weak solutions of mineral acids.

In this work, using multicomponent two-phase solvent systems, an experimental study on the extraction separation of REEs in columns with sieve plates operating in the liquid–liquid chromatography mode was conducted to assess the potential of this extraction equipment for large-scale chromatographic separation of rare earth elements. Compared with the CCC devices, these apparatuses are extremely simple and, as mentioned above, have a significantly higher productivity. In this study, for the first time, the possibility of operating a cascade of sieve plate extraction columns in the chromatography mode has been shown. For the first time, the chromatographic separation of REEs in the cascade of sieve plate extraction columns was experimentally studied under conditions of injections of large volumes of metal salt solution into the installation; it has been shown that this high-performance industrial equipment can be used for large-scale separation of rare earth elements. The main advantages of the separation of metals in a cascade of multi-stage extraction columns or in a cascade of mixer–settlers operating in the chromatographic mode over extraction methods are the separation of multicomponent mixtures within a single process stage (in one technological operation), which simplifies technological schemes; higher purity of the separated products; and low consumption of reagents and organic solvents, since they remain in the system. Thus, the application of chromatography mode can lead to significant cost savings.

**Multistage extraction columns operating in liquid–liquid chromatography mode**

There are two types of multistage extraction columns: (1) intensified columns, in which for the dispersion of one of the phases kinetic energy is supplied with the help of mixers (sectioned columns with various types of stirrers and pulsation columns); (2) spray columns with perforated trays, in which one of the phases is dispersed due to gravitational forces as it flows out of the tray holes. When operating in the countercurrent extraction mode, the columns have settling zones at the top and at the bottom for separation and withdrawal of the light and heavy phases. To operate in the conventional eluent chromatography mode, for the separation and withdrawal of the mobile phase it is sufficient to equip the columns with only one settling zone. Obviously, to carry out real countercurrent chromatography separations, the columns must be equipped with two settling zones.

Columns with sieve plates are the simplest type of extraction apparatus widely used in large-scale production of chemical products and petrochemistry. When using a cascade of these columns as an installation for liquid–liquid chromatography, depending on the design of the sieve plates and the physicochemical properties of the liquids, two hydrodynamic modes of separation processes are possible. In the first case, after crushing in the volume of the continuous stationary phase between adjacent trays, drops of the mobile phase coalesce with the formation of a continuous layer on the lower tray (when the mobile phase is the

heavy phase) or under the upper tray (when the mobile phase is the light phase). In the second case, the mobile phase is crushed at the top of the column (when the mobile phase is the heavy phase) or at the bottom of the column (when the mobile phase is the light phase); its drops pass through the entire column, and their coalescence occurs in the settling zone at the bottom or at the top of the column.

## 2. Materials and Methods

The experimental setup consisted of four sieve tray columns connected in series along the heavy mobile phase flow. Each column (from a $6.4 \times 9.6$ mm FEP tube) contained 25 PTFE plates 3 mm thick with 13 holes 0.25 mm in diameter spaced every 35 mm. The total volume of the setup was 119 mL; the volume of each stage in the columns was 1.13 mL; the number of stages in each column was 26; and the total number of stages in the setup was 104. The scheme of the experimental setup is shown in Figure 1.

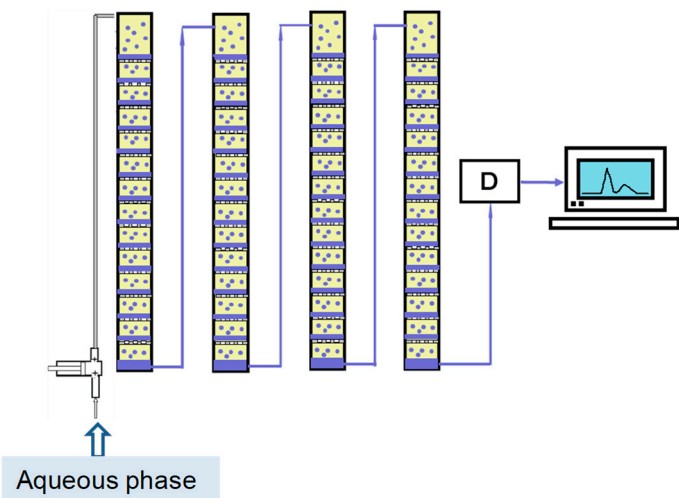

**Figure 1.** Mixing–settling mode of operation of a liquid–liquid chromatography unit, consisting of series-connected sieve plate columns.

The experimental setup operates as follows: the columns are filled with the light stationary extractant phase, and the heavy mobile phase is pumped through them with a piston or peristaltic pump.

The stock solutions of lanthanum(III) and neodymium(III) chlorides were prepared with dissolution of $LaCl_3 \cdot 7H_2O$ and $NdCl_3 \cdot 6H_2O$ salts (all chemically pure) in distilled water. The stock solutions of Sm(III), Eu(III), Er(III), and Tb(III) chlorides were obtained by dissolving weights of oxides of corresponding metals (all chemically pure) in concentrated HCl followed by evaporating the solutions in a water bath for removing excess acid.

The stock solutions of REE chlorides were used to prepare aqueous solutions of metal chlorides with addition of NaCl (0.1 M) and HCl (0.01 M). Cyanex 572 (a mixture of phosphonic and phosphinic acids) produced in Cytec was applied as an extractant. Hexane and isopropyl alcohol (all reagent grade) were applied as solvents. The feasibility of extraction–chromatographic separation of REEs in a cascade of extraction columns was experimentally tested. For this purpose, the chromatographic behavior of REE chlorides was studied in a 2-phase liquid system: 0.1 M Cyanex 572 in hexane–isopropyl alcohol—0.02 M aqueous solution of Ln(III) (0.01 M HCl and 0.1 M NaCl). The volume ratio of all components was 2:1:1.5.

After the establishment of a stable hydrodynamic regime, in which only the mobile phase leaves the system, an aqueous solution of metal salts was introduced into the upper stage of the first column using a syringe, and the relative content of metals in the mobile phase was continuously measured and recorded by the conductometric method at the outlet of the last column using a flow-through conductivity electrode connected to the PC.

## 3. Results and Discussion

Preliminary experiments established that in the selected multicomponent 2-phase solvent system, the retention of the stationary phase, depending on the mode of pumping through the columns of the mobile phase, varied by 80–90%. First, experiments were carried out using a piston pump, which provided a discrete supply of the mobile phase to the columns and the pulsating movement of liquids in them.

Individual chromatographic peaks of La(III) and Sm(III) chlorides and the chromatogram of a mixture of REE chlorides are shown in Figure 2.

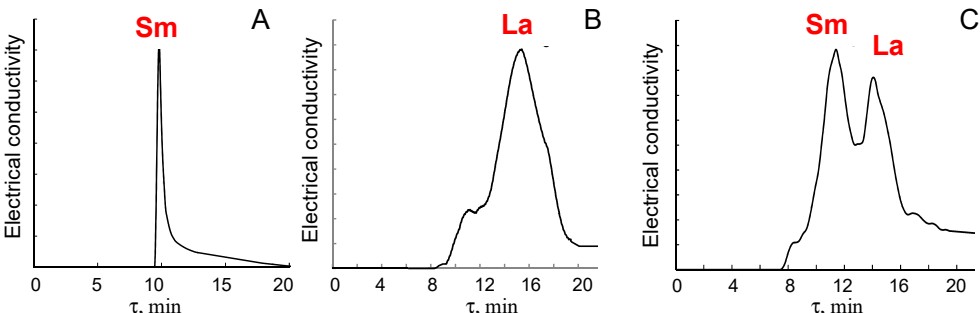

**Figure 2.** Experimental chromatographic peaks of SmCl$_3$ (**A**) and LaCl$_3$ (**B**) and the chromatogram of a mixture of LaCl$_3$ and SmCl$_3$ (**C**).

The unusual elution sequence of Sm and La may be explained by the following: in the multicomponent system, which involves hydrophilic and hydrophobic components, as well as a component with hydrophilic and hydrophobic properties (water–hexane–isopropyl alcohol), in addition to the chemical reaction between lanthanide ions and the Cyanex 572 extractant, the physical distribution of the extracted species between the phases proceeds. This difference between the distribution mechanisms in the multicomponent and extraction systems can affect the order of REE distribution in the multicomponent system.

The experimental chromatograms in Figure 2 confirm the possibility of separating metals in a cascade of series-connected sieve plate columns operating in the elution chromatography mode. The process of extracting metals is usually associated with a chemical reaction at the interface. Therefore, intensified multistage columns are to be used for the chromatographic separation of metals. The efficiency of metal separation in such devices can also be significantly increased by carrying out the process of extraction–chromatographic separation in a closed circulation loop. The sample components are repeatedly pumped through the columns when organizing the separation processes in the circulation loop; therefore, for the separation of metals according to the recirculation scheme, significantly fewer extraction stages will be required than in the scheme without recirculation. As an example, Figure 3 shows the experimental chromatogram of samarium obtained in the above column cascade after two sample passes.

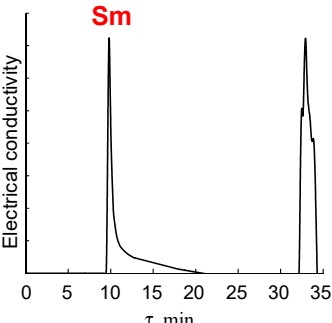

**Figure 3.** Experimental chromatographic peaks of SmCl$_3$ after two sample passes.

When certain conditions are met for introducing a solution of components into the cascade, it is possible to isolate concentrated fractions of the target components in the recycling process [24]. To model the considered methods of REE separation, the analytical dependences published in [22,24,25] can be used.

High-throughput chromatographic separation can be achieved by long and/or multiple sample loading. Figure 4 shows experimental chromatograms of binary mixtures of REE salts obtained in the mode of the continuous flow of the dispersed mobile phase (by using the peristaltic pump) under conditions of prolonged (A) and repeated (B) sample loading. In the first case (Figure 4A), due to the greater difference in the partition coefficients $K_D$, a more efficient separation of metals occurs. According to the obtained experimental results, the practical complete separation of lanthanum and erbium occurs within 60 min (Figure 4A), while the incomplete separation of lanthanum and samarium takes 20 min (Figure 2C). As above, this can be explained by differences in the partition coefficients $K_D$ of REEs. The values of $K_D$ calculated directly from the chromatographic peaks are $K_{DSm} = 0.05$; $K_{DLa} = 0.2$; $K_{DEr} = 0.55$; and $K_{DTb} = 0.42$.

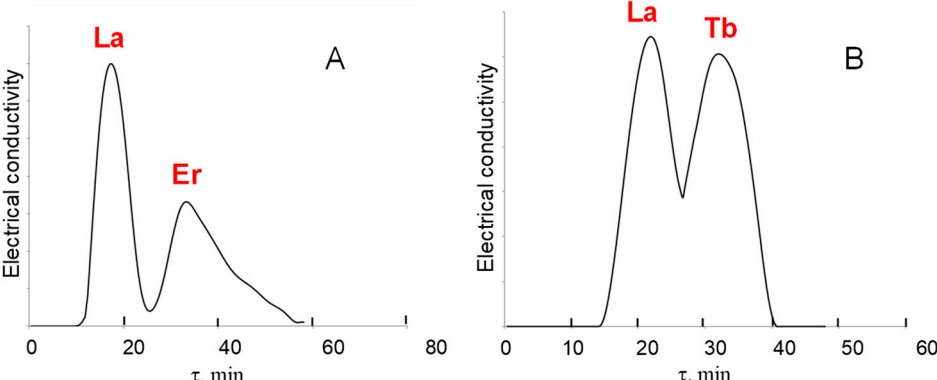

**Figure 4.** Experimental chromatograms of mixtures of REE chlorides at (**A**) long (sample loaded for 6 min) and (**B**) multiple sample loading (sample loaded 6 times within a minute with a 30 s interval for feeding the mobile phase).

Based on the above, we can propose the following technology for the chromatographic separation of rare earth elements: a continuous separation of REEs is conducted in a closed-loop recycling liquid–liquid chromatography mode, using a cascade of intensified multistage extraction columns; the recycling system includes a pump and connecting pipelines. At certain time intervals, the loop is opened and the solution of REEs is fed into the first column with the flow rate equal to the rate of the mobile phase circulation in the loop. After the loading of the REE solution, the loop is closed, and during a certain number of cycles the metals are separated. Then, the loop is opened again; an appropriate mobile phase is fed into the system with the flow rate equal to the rate of the mobile phase circulation in the loop, and the separated metal fractions are removed from the cascade. The supply of the feed solution and the removal of the metal fractions can be carried out at different times.

To evaluate the applicability of the proposed technology in the separation of rare earth elements, computer simulation of these processes was carried out.

## 4. Simulation of Chromatographic Separation of Rare Earth Elements by Closed-Loop Recycling Liquid–Liquid Chromatography

In liquid–liquid chromatography, as in solvent extraction, the separation of compounds is influenced by interphase mass transfer and longitudinal dispersion of the compounds in a chromatographic column caused by axial mixing in stationary and mobile phases. To describe the elution profile of compounds in chromatographic devices and to interpret the experimental results, two models were used: the continuous diffusion model and staged cell model [37–39]. Both models take into account two band-broadening mechanisms:

interfacial mass transfer and longitudinal mixing due to the uneven flow velocity profile of the mobile phase, turbulence, and molecular diffusion. For low degrees of longitudinal mixing, as is the case in chromatographic devices, simulations by continuous and staged models give practically identical results. However, the mathematical description of chromatography by the cell model, which represents a chain of ideally mixed, equally sized tanks (cells), is much simpler. In the cell model, the effects of interfacial mass transfer and longitudinal mixing is characterized by one parameter—the number of cells $N$:

$$N = \frac{nT(1+K')^2}{T(1+K')^2 + 2nK'^2} \tag{1}$$

Equation (1) determines the contribution of longitudinal mixing, characterized by the number of ideally mixed cells $n$, and interfacial mass transfer, characterized by the number of transfer units $T = a_c\,k_x\,V_c/F$, to the efficiency of a separation process in liquid–liquid chromatography.

In Equation (1), $K' = \frac{K_D S_f}{1-S_f}$ is the dimensionless capacity factor; $K_D = y/x$ is the distribution coefficient; $x$ is the concentration in the mobile phase; $y$ is the concentration in the stationary phase; $S_f$ is the fraction of column volume ($V_c$) occupied by the stationary phase; $a_c$ is the specific contact surface of the phases; $k_x$ is the mass transfer coefficient; and $F$ is the volumetric flow rate of the mobile phase.

Replacing the number of theoretical plates in the known equilibrium chromatography model by the number of theoretical cells defined by Equation (1) makes it possible to use a simple model of theoretical plates (or equilibrium cells) in the analysis of non-equilibrium processes of liquid–liquid chromatography.

Based on the cell model and the results of our previous studies [16,25,39–42], the following equations can be developed to simulate the separation of a binary mixture of rare earth elements by liquid–liquid chromatography in the conventional elution mode:

$$X(t) = \frac{a\sqrt{6N}}{\sqrt{\pi(Na^2t_s^2 + 12)}}exp\left[-\frac{3N(2 + at_s - 2at)^2}{2(Na^2t_s^2 + 12)}\right] \tag{2}$$

$$X_{1,2}(t) = q_1 X_1(t) + q_2 X_2(t) \tag{3}$$

Equation (2) describes the elution profile of individual rare earth elements, and Equation (3) describes the chromatogram of a mixture of two rare earth elements after the loading of a solution, containing $q_1$ and $q_2 = 1 - q_1$ proportions of the first and second rare earth elements. When developing Equation (3), it was assumed that there is no significant interaction between the components of the mixture.

In Equations (2) and (3), $a = \frac{1}{1-S_f+S_f K_D}$; $q_1 = Q_1/Q$, $q_2 = Q_2/Q$: $Q_1$ and $Q_2$ are the amounts of the first and the second rare earth elements in a loaded solution of rare earth elements; $Q = Q_1 + Q_2$ is the total amount of rare earth elements in the loaded solution; $X = x/\bar{x}$ is the dimensionless (normalized) concentration of individual rare earth elements; $\bar{x} = x_s F \tau_s / V_c$ is the mean concentration in the chromatographic device after the loading of the rare earth element solution; $X_{1,2}$ is the normalized total concentration of the metals; $t_s = \tau_s F / V_c$ is the normalized rare earth element solution loading time; $x_s$ is the concentration of a rare earth element in the feed; and $\tau_s$ is the rare earth element solution loading time.

The results of the simulation of the separation of the La/Er mixture by Equations (2) and (3) are shown in Figure 5. The process parameters $N = 40$; $K_{DLa} = 0.2$; and $K_{DEr} = 0.55$ were determined from the experimental chromatograms in Figure 4. The results in Figures 4 and 5 demonstrate an acceptable agreement between the experiment and the mathematical model presented above.

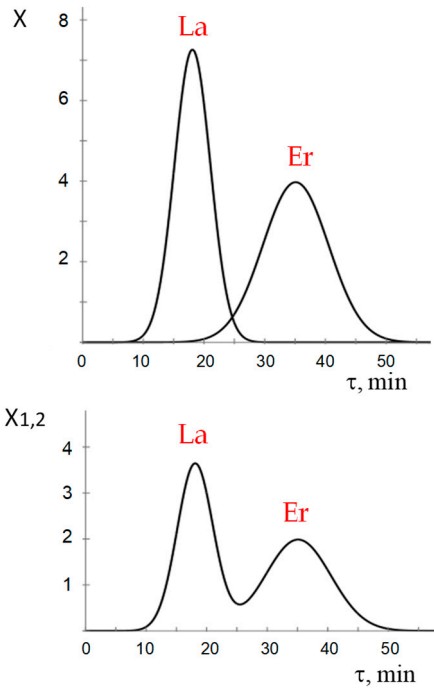

**Figure 5.** Simulation of the separation of La/Er mixture by Equations (2) and (3). Process parameters: $N = 40$; $K_{DLa} = 0.2$; $K_{DEr} = 0.55$; $S_f = 0.9$; $q_1 = 0.5$.

In liquid–liquid chromatography devices, the high-volume fraction of the stationary phase enhances the separation of components of a mixture. However, the separation efficiency of liquid–liquid chromatography (LLC) is much lower than that of HPLC. The efficiency of LLC separations can be significantly improved by using elution schemes and modes that simulate the increase in the number of theoretical cells (plates) of the LLC system, such as dual mode, multiple dual mode [26,39], and closed-loop recycling mode. The closed-loop recycling LLC method is the simplest way to increase the number of theoretical plates of LLC plants. It provides significant solvent savings and requires no additional equipment. In this method, the sample or certain parts of the elution profile are recycled several times in the LLC system until the required degree of separation of the components is reached. With an increase in the number of sample passages through the plant (in the number of cycles), the quality of the separation improves due to the increase in the number of theoretical plates. However, at the same time, the chromatograms of neighboring cycles begin to converge: components with low distribution coefficients of the current cycle catch up with components with high distribution coefficients from the previous cycle, and, after a certain number of cycles, the chromatograms of neighboring cycles begin to overlap. To take into account these counteracting phenomena, two approaches are used [40–42]: (1) When the volume of the recycling system consisting of connecting lines, valves, a pump, a detector, etc., does not exceed one percent of the volume of the LLC plant, the effects of extra-column dispersion on the separation can be neglected [40]. (2) The recycling system, which allows for the recycling of the mobile phase through the LLC plant, is included in the mathematical model, replacing it with a cascade of ideally mixed cells. Thus, there are two options to design an LLC installation operating in closed-loop recycling mode: (1) with short recycling tubing (small volume of the recycling system compared with the LLC device); (2) with long recycling tubing (with a certain volume of the recycling system). In a recycling system with long, small-diameter tubing, where a large volume of the recycling system can be provided due to the length of the connecting pipes, the flow of the mobile phase approaches the ideal displacement mode, which allows an increase in the number of sample passages through the LLC device and therefore improves separation. In this case,

the following equation can be used to simulate the separation of rare earth elements by closed-loop recycling liquid–liquid chromatography:

$$X_m(t) = \sum_{i=1}^{m} \frac{a\sqrt{12N}}{\sqrt{2\pi(Na^2t_s^2 + 12i)}} exp\left[-\frac{3N(2i + at_s + 2ab(i-1) - 2at)^2}{2(Na^2t_s^2 + 12i)}\right] \tag{4}$$

In Equation (4), $b$ is the ratio of the recycling system volume and the LLC device volume; $m$ is the number of cycles. Equation (4) describes the change in the concentration profiles of individual rare earth elements at the outlet of the installation for the entire circulation time from the first to the last cycle $m$. Figure 6 shows the results of the simulation of the separation of samarium ($K_{DSm} = 0.05$), lanthanum ($K_{DLa} = 0.2$), and erbium ($K_{DEr} = 0.55$) in the liquid–liquid chromatography plant with $N = 50$ and $b = 0.8$ operating in the closed-loop recycling mode. Figure 6 shows the chromatograms after the first and second cycles. As seen, an acceptable separation of the metals can be reached after two cycles.

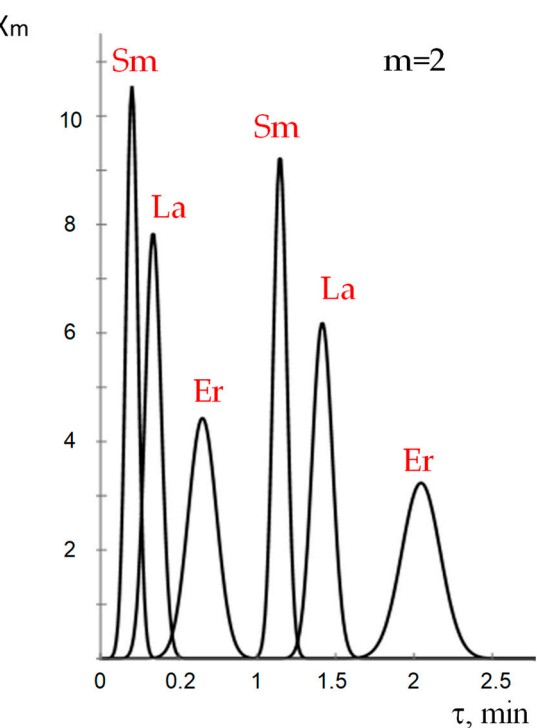

**Figure 6.** Simulation of the separation of samarium ($K_{DSm} = 0.05$), lanthanum ($K_{DLa} = 0.2$), and erbium ($K_{DEr} = 0.55$) in a liquid–liquid chromatography plant operating in the closed-loop recycling mode. Process parameters: $N = 50$; $S_f = 0.9$; and $b = 0.8$.

## 5. Conclusions

In this study, based on a cascade of multistage extraction columns, a chromatographic technology is proposed for the separation of REEs.

The chromatographic separation of REEs in the cascade of sieve plate extraction columns was experimentally studied under conditions of single and multiple injections of large volumes of metal salt solution into the installation. The feasibility of operating the cascade of sieve plate extraction columns in the chromatography mode was demonstrated.

For the separation of rare earth elements, a method of closed-loop recycling liquid–liquid chromatography is proposed. Computer simulation of the chromatographic separation of rare earth elements by closed-loop recycling liquid–liquid chromatography showed the feasibility and potential of this method. Industrial separation of rare earth elements by closed-loop liquid–liquid chromatography can be implemented on the basis of a cascade of

multistage extraction columns or a cascade of mixer–settler extractors with semi-continuous loading of the solution of the metals to be separated.

For the further development and industrial implementation of the considered technology, experimental studies should be conducted using intensified multistage extraction columns, such as sectioned columns with agitators and vibrating plate columns.

**Author Contributions:** Conceptualization, A.E.K.; methodology, A.E.K. and V.V.B.; validation, V.V.B. and Y.V.T.; formal analysis, A.E.K. and V.V.B.; writing, review, and editing, A.E.K. and V.V.B.; visualization, V.V.B., Y.V.T. and M.M.P.; project administration, V.V.B. All authors have read and agreed to the published version of the manuscript.

**Funding:** This research was funded by the Russian Science Foundation, grant number 23-29-00162.

**Institutional Review Board Statement:** Not applicable.

**Informed Consent Statement:** Not applicable.

**Data Availability Statement:** Not applicable.

**Conflicts of Interest:** The authors declare no conflict of interest.

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
