# Peer review of "Separation of Rare Earth Elements in Multistage Extraction Columns in Chromatography Mode: Experimental Study and Mathematical Simulation"

_processes, doi:10.3390/pr11061757_

Round 1

Reviewer 1 Report

This paper is related to the application of chromatography to the REEs mutual separation. This is also as an application of solvent extraction used in hydrometallurgy. It seems that the authors aim to develop such a technique which can be utilized in the actual REEs production.

However, the presented experimental data are still in the initial stage. Mutual separation data of La and Er (Fig. 5 A) and La and Tb (Fig. 5B) are shown. But in conventional solvent extraction using acidic organophosphorus reagent it is not difficult to achieve these mutual separations. Thus, it does not seem that the data in Fig. 5 have sufficient impact on the community of hydrometallurgy.

In the last paragraph of Section 3, the authors propose a method for mutual separation. But this is just a proposal, and no experimental validation is shown.

Thus, I cannot think that this paper is worth publishing in this journal.

Reviewer 2 Report

The work presents the possibility of the application of the chromatographic separation of rare earth elements. The paper demonstrates preliminary studies and can be treated as a technical note rather than a regular research paper.

The major concerns about the work include:

1. The novelty of the work should be exposed better.

2. The proposed procedure should be examined with various multicomponent mixtures.

3. More quantitative data need to be presented.

Reviewer 3 Report

In the manuscript, the authors evaluated the use of sieve plate extraction columns in the liquid-liquid chromatography mode to separate rare earth elements. Successful separation of the rare earth elements from model solutions was demonstrated with intensified multi-stage columns. The topic is of high research interest. However, the significance of the study is not highlighted, nor are the results clearly presented. The followings are my comments:

1. The introduction does not include sufficient background information. Why do we want to separate rare earth elements and what are the difficulties? What are the traditional techniques for rare earth element separation and what are their issues? What are the difficulties of scaling-up liquid-liquid chromatography?

2. The authors stated the disadvantages of the CCC processes to be the complexity of instrumentation and low productivity, but believed sieve plate extraction columns have the potential for large-scale operation. Then it is necessary to report how the sieve plate extraction columns affect the instrumentation complexity and productivity.

3. The mechanism and background information of multistage liquid-liquid chromatography is not clearly stated. It might be helpful to include some figures and process diagrams as visual aids to help readers better understand.

4. Please report the concentrations of the model solutions used in the study and justify them. In other words, which source/applications are they prepared to simulate (i.e., seawater, inland groundwater, ...)?

5. In the results section, please report the rare earth elements' concentrations/purities before and after separation. Please also report the process productivity.

6. Please emphasize in the last paragraph of the introduction the novelty and significance (scientific contribution and influence) of the current study.

Round 2

Reviewer 1 Report

I think that the publication of this manuscript should be done after their proposed method is validated by real experiment or, at least, computer simulation by some reliable quantitative model. At best, as the reviewer #2 wrote, “technical note” is suitable. Anyway, I respect the editor’s decision.

When this manuscript is published in any category, I point out the following to help the readers’ understanding.

(1)  The title is not appropriate. In this manuscript, the authors are studying the separation of REEs. This aspect should be included in the title.

(2)  The authors emphasize the application of chromatographic method to hydrometallurgy. In hydrometallurgy, mass production at lower cost is needed. Also, when using solvent extraction, lower solvent inventory is favorable. In these aspects, is the proposed method better than the conventional mixer-settler method? They should compare these points to convince the readers. Otherwise, the sentence in L81-82 seems to be dubious. Also, in Introduction, the problems of the conventional mixer-settler method should be described.

(3)  L142-143: The concentrations of the metals were NOT measured. They just measured electrical conductivity of the aqueous solution.

(4)  Figs. 2-3: They should explain the reason for the sequence of time when the peak appeared. From the figures, the sequence is Eu < Sm < Nd < La; that is, the lighter RE needs more time. It is known that the affinity of phosphonic and phosphinic acid extractants to REEs increases with the atomic number. Thus, the heavier RE would need more time.

(5)  Figures 2A, 3B: There are shoulders before and after the main peaks. If possible, please explain the reasons.

(6)  L190-193: “Fidure” should be “Figure”.

(7)  Fig. 5: The sequence of the peaks is reversed from those of Figs. 2-3. They should explain the reason.

Reviewer 3 Report

The authors have addressed most of my comments well, except for comments 4 and 5. 

4: "Please report the concentrations of the model solutions used in the study and justify them. In other words, which source/applications are they prepared to simulate (i.e., seawater, inland groundwater, ...)?"

- When a mock aqueous solution was used (0.02 M aqueous solution of Ln(III), 0.01 M HCl and 0.1 M NaCl), the authors need to justify the selected composition. Why 0.02 M Ln(III) was used in the feed solution?

5: "In the results section, please report the rare earth elements' concentrations/purities before and after separation. Please also report the process productivity."

- Reporting only separation time is insufficient. Need to also report the volume/mass of the separated elements (or processed feed aqueous solution) per unit of time. 

- Still need to report the purity/concentration of the element after separation. In other words, are you able to increase the targeted element purity after separation?

Additional comments:

1. Need transition sentence:

after "In addition to extraction methods, crystallization, adsorption and chromatography methods are used for REE separation." (lines 28-29).

between "CCC devices ... are difficult to scale up" (lines 43-46) and "The disadvantages of CCC ..." (lines 46-49).

2. Please report a quantitative range of productivity for the industrial extraction apparatus.

3. Do you need any regeneration steps to recycle the solvent?

Round 3

Reviewer 1 Report

The authors have added the simulation part and validated their idea of closed-loop recycling mode. I have only one comment as below.

In Fig. 2, t(Sm) < t(La). But, as I pointed out in my former review, the sequence should be t(Sm) > t(La), because Sm3+ should have higher affinity to Cyanex 572 than La3+. They did not answer this point in the former reply. Even if the conditions of Fig. 2A and Fig. 2B are different, this is not the case in Fig. 2C because they used mixed solution of La and Sm. I am afraid that Fig. 2A is for La and Fig. 2B for Sm. Also, in Fig. 2C, the left peak is for La and the right peak is for Sm. I recommend the authors to check it. Accordingly, the simulation part can be corrected. If the sequence of t (t(Sm) < t(La)) or KD (KDSm < KDLa) is correct, they should explain the reason.

Reviewer 3 Report

The revised version of the manuscript is overall satisfactory. Minor editing of the English language will be required. It is also unclear that in Figure 3, are both peaks Sm?

Author Response

Please see tha attachment.
